

**Recent changes and drivers of the atmospheric evaporative demand in the Canary**
**Islands**
Vicente-Serrano, S.M.[1], Azorin-Molina, C.[1], Sanchez-Lorenzo, A.[1], El Kenawy, A.[2], Martín-
Hernández, N.[1], Peña-Gallardo, M.[1], Beguería, S.[3], Tomas-Burguera, M.[3]
*[1]Instituto Pirenaico de Ecología, Consejo Superior de Investigaciones Científicas (IPE–CSIC), Zaragoza, Spain;*
*[2]Department of Geography, Mansoura University, Mansoura, Egypt;[3]Estación Experimental Aula Dei,*
*Consejo Superior de Investigaciones Científicas (EEAD-CSIC), Zaragoza, Spain.*
* Corresponding author: svicen@ipe.csic.es
**Abstract**
We analysed recent evolution and meteorological drivers of the atmospheric evaporative demand
(AED) in the Canary Islands for the period 1961 -2013. We employed long and high quality time
series of meteorological variables to analyze current AED changes in this region and found that
AED has increased during the investigated period. Overall, the annual ETo increased significantly
by 18.2 mm decade$^{-1}$ on average, with a stronger trend in summer (6.7 mm decade$^{-1}$). The radiative
component showed much lower temporal variability than the aerodynamic component did. Thus,
more than 90% of the observed ETo variability at the seasonal and annual scales can be associated
with the variability of the aerodynamic component. The variable that recorded more significant
changes in the Canary Islands was relative humidity, and among the different meteorological factors
used to calculate ETo, relative humidity was the main driver of the observed ETo trends. The
observed trend could have negative consequences in a number of water-depending sectors if it
continues in the future.
**Key-words:** Reference Evapotranspiration, Aerodynamic component, Radiative component,
Temporal changes, Potential Evapotranspiration, Global warming, Canary Islands.



## 1. Introduction

The atmospheric evaporative demand (AED) is one of the key variables of the hydrological cycle (Wang and Dickinson, 2012), with multiple implications for agriculture, hydrology and the environment (Allen et al., 2015). Several studies have indicated that current global warming is increasing the intensity of the hydrological cycle, mainly as a consequence of an intensification of the AED (Huntington, 2006). Sherwood and Fu (2014) suggested that mechanisms driving the AED over land regions could be the main driver of increasing climate aridity in world semi-arid regions under a global warming scenario.

Warming may play an important role in increasing the AED via the aerodynamic component (McVicar et al., 2012a). Following the Clausius-Clapeyron relationship, the quantity of water vapour that a given mass of air can store increases exponentially with the air temperature. Nevertheless, there are other climate variables whose temporal evolution could compensate the increased AED induced by increasing air temperature, such as wind speed and vapour pressure deficit (McVicar et al., 2012a). In addition, the radiative component of the AED, which is related to the available solar energy that transforms a unit of liquid water into vapour, may compensate or accentuate the increase in AED associated with warming. Wild et al. (2015) noted that solar radiation increased over large regions since the 1980s as a consequence of changes in cloud cover and/or atmospheric aerosol concentrations.

These large number of variables interact in a non-linear manner to determine the AED (McMahon et al., 2013), so assessing recent changes in the AED and defining their determinant factors is not an easy task. For this reason, while several studies analysed the AED at the global scale using different datasets and methods, there is no general consensus on the recent AED evolution (Sheffield et al., 2012; Matsoukas et al., 2011; Wang et al., 2012; Dai, 2013). In this context, the few existing direct AED observations, based on evaporation pans, show a decrease since the 1950s at the global scale (Peterson et al. 1995; Roderick and Farquhar 2002 and 2004), a finding that adds more uncertainty



regarding the behaviour of the AED under current global warming. These issues stress the need for
new studies that employ high quality datasets to assess the time evolution of the AED at the
regional scale.
There are a number of studies published in the last decade that analysed the AED evolution across
different regions of the World. Some of them are based on AED estimated using empirical
formulations, mostly based on air temperature data (e.g., Thornthwaite, 1958; Hargreaves and
Samani, 1995). However, to adequately quantify the AED evolution it is necessary to use long-time
series of the meteorological variables that control its radiative and aerodynamic components (e.g.
air temperature, vapour pressure deficit and wind speed). Although these variables are generally
poorly measured and highly inhomogeneous over both space and time, numerous regional studies
analysed the evolution of the AED by means of the robust Penman-Monteith (PM) equation using
long times series of these variables. The available regional studies show quite contradictory results,
where some studies showed AED negative trends, including those in China (Xu et al., 2006; Ma et
al., 2012; Zhang et al., 2007; Liu et al., 2015) and northwest India (Jhajharia et al., 2014). In
contrast, other regional studies found positive trends in AED, including those in central India
(Darshana et al., 2012), Iran (Kousari and Ahani, 2012; Tabari et al., 2012), Florida (Abtew et al.,
2011), continental Spain (Espadafor et al., 2011; Vicente-Serrano et al., 2014a; Azorin-Molina et
al., 2015), France (Chaouche et al., 2010) and Moldova (Piticar et al., 2015).
The contrasted trends among world regions would be a consequence of the evolution of the
different meteorological variables that control the AED. Specifically, some studies suggest that
temporal variability and changes in the AED are related to changes in the relative humidity, mainly
in semi-arid regions (Wang et al., 2012; Espadafor et al., 2011; Vicente-Serrano et al., 2014b),
whereas others stress the importance of solar radiation (Roderick and Farquhar, 2002; Roderick et
al., 2007; Ambas and Baltas, 2012; Fan and Thomas, 2013) or wind speed (McVicar et al., 2012b).




Among these studies, few analyzed the AED variability and trends and their possible drivers  in the
eastern North Atlantic region (Chaouche et al., 2010; Vicente-Serrano et al., 2014a; Azorin-Molina
et al., 2015). Nevertheless, there are no studies about this issue in the sub-tropical areas of the north
Atlantic region. In this area, there are very few meteorological stations measuring long-term series
of the variables necessary to make robust calculations of the AED. This uneven distribution of
meteorological observatories constrains the high interest to know the evolution of atmospheric
processes in this region, where climate variability is strongly controlled by changes in the Hadley
circulation (Hansen et al., 2005) that affects the position and intensity of the subtropical anticyclone
belt. Knowing the evolution of AED and its main drivers in this region is highly relevant given the
general climate aridity of the region and the low availability of water resources (Custodio and
Cabrera, 2002). In this work we analyze the recent evolution and meteorological drivers of the AED
in the Canary Islands. The availability of long and high quality time series of meteorological
variables in the Canary Islands provides an opportunity to analyze current AED changes in the sub-
tropical northeastern Atlantic region and the role played by different meteorological variables.

**2. Methods**
***2.1. Dataset***
We used the complete meteorological records of the Spanish National Meteorological Agency
(AEMET) in the Canary Islands for the following variables at the monthly scale: maximum and
minimum air temperature (308 stations), wind speed (99), sunshine duration (42) and mean relative
humidity (139). A majority of the stations cover short periods or are affected by large data gaps. As
the number of meteorological stations before 1961 was very little for several variables we restricted
our analysis to the period between 1961 and 2013. Specifically, only 8 meteorological stations had
data gaps of less than 20% of the months in all the necessary variables. As illustrated in Figure 1,
these stations are distributed between the Islands of Tenerife (3 stations), Gran Canaria (2), La



Palma (1), Lanzarote (1) and Fuerteventura (1). Given that some series included records for a longer
period (e.g., Izaña from 1933 and Santa Cruz de Tenerife from 1943), neighbouring stations with
shorter temporal coverage were used to reconstruct the existing data gaps in the selected
observatories, using a regression-based approach.
Then, the time series were subject to quality control and homogenization procedures. The quality
control procedure was based on comparison of the rank of each data record with the average rank of
the data recorded at adjacent stations (Vicente-Serrano et al., 2010). A relative homogeneity method
was applied to identify possible inhomogeneities. For this purpose, we used HOMER
(HOMogenization software in R), which compares each candidate series with a number of available
series (Mestre et al., 2013). The method provides an estimation of break points in the time series
relative to other stations, indicating high probabilities of the presence of inhomogeneities. This
method was applied to the different variables and time series following Mestre et al. (2013). Finally,
a single regional series for the different variables was obtained using a simple arithmetic average of
data values at the available eight stations.

*2.2. Calculation of ETo*
The Penman-Monteith equation (PM) equation is the standard technique for calculation of ETo
from climatic data (Allen et al.,1998), and it is the method officially adopted (with small variations)
by the International Commission for Irrigation (ICID), the Food and Agriculture Organization
(FAO) of the United Nations, and the American Society of Civil Engineers (ASCE). The PM
method can be used globally, and has been widely verified based on lysimeter data from diverse
climatic regions (Allen et al., 1994; Itenfisu et al., 2000; López-Urrea et al., 2006). Allen et al.
(1998) simplified the PM equation, developing the FAO-56 PM equation, and defined the reference
surface as a hypothetical crop with assumed height of 0.12 m, surface resistance of 70 s m$^{-1}$ and
albedo of 0.23 that had evaporation similar to that of an extended surface of green grass of uniform
height that was actively growing and adequately watered. The ETo FAO-56 PM is expressed as:
$$ET_o = \frac{0.408 \cdot \Delta \cdot (R_n - G) + \gamma \cdot \frac{900}{T + 273} \cdot u_2 \cdot (e_s - e_a)}{\Delta + \gamma \cdot (1 + 0.34u_2)}$$
(1)


where $ETo$ is the reference evapotranspiration (mm day$^{-1}$), $R_n$ is the net radiation at the crop surface
(MJ m$^{-2}$ day$^{-1}$), $G$ is the soil heat flux density (MJ m$^{-2}$ day$^{-1}$), $T$ is the mean air temperature at 2 m
height (ºC), $u_2$ is the wind speed at 2 m height (m s$^{-1}$), $e_s$ is the saturation vapour pressure (kPa), $e_a$
is the actual vapour pressure (kPa), $e_s$-$e_a$ is the saturation vapour pressure deficit (kPa), $\Delta$ is the
slope of the vapour pressure curve (kPa ºC$^{-1}$), and $\gamma$ is the psychrometric constant (kPa ºC$^{-1}$). Thus,
the monthly ETo can be calculated from data of the monthly averages of five meteorological
parameters: maximum and minimum air temperature, relative humidity (which allows calculating
the vapour pressure deficit), wind speed at a height of 2 m, and daily sunshine duration (which
allows estimating the net radiation). Further details on the required equations to obtain the
necessary parameters from meteorological data can be consulted in Allen et al. (1998).
We also calculated the evolution of the radiative (Eq.2) and the aerodynamic (Eq.3) components of
the ETo, as follows:
$$ETo(r) = \frac{[0.408\Delta(Rn - G)]}{[\Delta + \gamma(1 + 0.34u_s)]}$$
(2)

$$ETo(a) = \frac{\left[\gamma\left(\frac{900}{T + 272}\right)u_2(e_s - e_a)\right]}{[\Delta + \gamma(1 + 0.34u_s)]}$$
(3)


***2.3. Analysis***
Using the time series of ETo, we determined the seasonal (winter: December–February; spring:
March–May; summer: June–August; autumn: September–November) and annual ETo averages. To
analyze changes in ETo we used the nonparametric Mann-Kendall statistics that measures the





degree to which a trend is consistently increasing or decreasing. The Mann-Kendall statistic is
advantageous compared to parametric tests as it is robust to outliers and it does not assume any
underlying probability distribution of the data (Zhang et al., 2001). For these reasons, it has been
widely used for trend detection in a wide range of hydrological and climatological studies (e.g.,
Zhang et al., 2001; El Kenawy and McCabe, 2015). To assess the magnitude of change, we used a
linear regression analysis between the series of time (independent variable) and the ETo series
(dependent variable). The slope of the regression indicated the amount of change (ETo change per
year), with higher slope values indicating greater change. We also calculated the trend observed in
the different meteorological variables (air temperature, relative humidity, sunshine duration and
wind speed) at both the seasonal and annual scales.
To get insight into the influence of changes in the different meteorological variables on ETo, we
related the evolution of ETo with relative humidity, maximum and minimum air temperature, wind
speed and sunshine duration by means of correlation analyses. To assess the importance of trends in
the different meteorological variables on the observed trends in ETo between 1961 and 2013, we
applied the PM equation while holding one variable as stationary (using the average from 1961 to
2013) each time. This approach provided five simulated series of ETo, one per input variable, which
could be compared to the ETo series computed with all the data to determine the isolated influence
of the five variables. Significant differences between each pair of ETo series (the original one and
the alternative one in which one variable was kept constant) were assessed by comparing the slopes
of the linear models, with time as the independent variable. A statistical test for the equality of
regression coefficients was used (Paternoster et al., 1998). The significance of the difference was
assessed at a confidence interval of 95% ($p<0.05$).

**3. Results**
**3.1. Average ETo values**





Figure 2 shows a box-plot with the seasonal and annual values of ETo in the different
meteorological stations across the Canary Islands, which are also summarized in Table 1. There
were strong seasonal differences in ETo, as all different meteorological stations show their
maximum values in summer and minimum in winter, albeit with strong differences among them. In
winter, the highest average values were recorded in the most arid islands (i.e., Fuerteventura and
Lanzarote) and in the station of Los Rodeos (North Tenerife). In summer, the stations of Izaña and
Los Rodeos showed the highest average values (663.8 and 612.9 mm, respectively). The lowest
summer ETo averages were recorded at the stations of Gran Canaria island (San Cristóbal and Gran
Canaria/Airport). At the annual scale, there were very few differences in the average values
between the stations of Los Rodeos, Izaña, Fuerteventura and Lanzarote, with very high ETo values
ranging between 1693 and 1784 mm (Table 1). The observatory with the lowest ETo values is
located in Gran Canaria Airport, although the observatory of San Cristóbal (also in the Gran
Canaria island) records the minimum values in summer. The magnitude of the differences can be
quite important (up to 34%) between the highest ETo values recorded in Los Rodeos, Izaña,
Fuerteventura and Lanzarote and the lowest ETo values (Gran Canaria and San Cristóbal). In
general, variability, as revealed by the coefficient of variation, was higher in the meteorological
stations that recorded the highest ETo values at the annual scale, but there was no clear spatial
pattern at the seasonal scale as different stations showed few differences in terms of the coefficients
of variation (Table 1).
In the majority of weather stations the seasonal and annual ETo magnitude was mostly driven by
the aerodynamic component. The average aerodynamic fraction was higher than the radiative
fraction in the weather stations that record the highest ETo values (Los Rodeos and Izaña) in all
seasons around the year (Figure 3). In other weather stations (Sta. Cruz de Tenerife and San
Cristóbal), the ETo associated with the radiative component was much higher than that observed for
the aerodynamic component (Table 2). The temporal variability in the aerodynamic component was





much higher than that observed in the radiative one, regardless of the season of the year or the
meteorological station.

**3.2. Long-term evolution of ETo**

The regional ETo series for the whole Canary Islands (Figure 4) shows a significant increase at the
annual scale (18.2 mm decade$^{-1}$), which is stronger in summer (6.7 mm decade$^{-1}$) (Table 3).
Nevertheless, there was a strong variability between the different meteorological stations, since
most meteorological stations experimented significant increases of ETo between 1961 and 2013.
The largest annual increase was recorded in Los Rodeos (34.8 mm decade$^{-1}$), La Palma (29.8 mm
decade$^{-1}$) and Lanzarote (29.7 mm decade$^{-1}$). Considering a longer period (1933-2013 for Izaña, and
1943-2013 for Santa Cruz de Tenerife), the changes are not statistically significant, although it was
not possible to check the homogeneity of the climate records prior to 1961 and thus the results for
the longer period must be carefully considered. For the period 1961-2013, there is no general spatial
pattern in the observed changes, thus some differences can be observed. For example, in the Gran
Canaria island, San Cristóbal station shows a statistically non-significant negative change in ETo on
the order of -8.4 mm decade$^{-1}$, while there is a general significant increase of 28.4 mm decade$^{-1}$ in
the Gran Canaria Airport.
Trends in the aerodynamic and radiative components showed clear differences among stations and
for the average Canary Islands (Figure 5). Main changes were recorded in the aerodynamic
component. The regional series showed an increase of 16.2 mm decade$^{-1}$ in the aerodynamic
component, but it only showed an increase of 2 mm decade$^{-1}$ in the radiative component (Table 4).
This can be translated to an average increase in the ETo of 89% over the whole period due to
changes in the aerodynamic component, and of 11% due to changes in the radiative component.
However, there are spatial differences between the meteorological stations, since the aerodynamic
component showed a decrease of 21 mm decade$^{-1}$ in San Cristóbal, compared to an increase of 44.6





mm decade$^{-1}$ in Los Rodeos. On the contrary, the radiative component showed lower differences
among stations, with values ranging from -9.9 mm decade$^{-1}$ in Los Rodeos to 12.7 mm decade$^{-1}$ in
San Cristóbal. Nevertheless, and regardless of the observed trends, the results indicate that the inter-
annual variability of ETo between 1961 and 2013 was mainly driven by the aerodynamic
component, independently of the season or the meteorological station considered (Table 5). The
temporal correlation between ETo and the aerodynamic component was statistically significant for
the different meteorological stations in the seasonal and the annual series, with correlation
coefficients higher than 0.95 in most cases. The correlation for the regional series was also strong
and statistically significant. In contrast, the correlation coefficients calculated between ETo and the
radiative component were much lower, and generally non-significant ($p<0.05$). Los Rodeos is the
unique weather station where the correlation between ETo and the radiative component was
statistically significant at both the seasonal and annual scales, but showing a negative correlation.
Overall, the results show that the correlation between the annual radiative component and the total
annual regional series of ETo is statistically non-significant.

**3.3. Drivers of ETo variability and trends**
Table 6 shows the correlation between the different meteorological variables and ETo at the
seasonal and annual scales in the eight meteorological stations. Maximum and minimum air
temperatures were positively correlated with ETo and this relationship was statistically significant
in some stations, and the correlation coefficients tended to be higher for maximum air temperature.
In Los Rodeos and La Palma, the ETo variability could not be explained by the variability in air
temperature, with correlation coefficients weaker than 0.3. Overall, the results indicate that the
seasonal and annual series of ETo were significantly correlated with variations of sunshine duration
and wind speed, suggesting that these two variables are the key drivers of ETo variability in the
Canary Islands. The variable that showed the strongest correlation with the evolution of ETo in the



seasonal and annual series of the different meteorological observatories was relative humidity, with
negative coefficients. Only in the annual series of Santa Cruz de Tenerife the correlation was non-
significant. Moreover, there were no significant differences in the magnitude of correlations among
seasons.
The regional series summarise the pattern observed in the individual meteorological stations (Figure
6). In winter, relative humidity had the strongest correlation with ETo (r=-0.85), with a mostly
linear relationship. Minimum air temperature and sunshine duration showed significant positive
correlations with ETo (r=0.40 and 0.36, respectively). Maximum air temperature and wind speed
showed weaker correlation with the winter ETo. In spring, the magnitude of the correlations was
similar among the different variables, and the highest correlation corresponded again to relative
humidity (r=-0.72). A similar pattern was found in summer, where relative humidity showed the
strongest correlation (r=-0.74) followed by maximum and minimum air temperature. In autumn,
relative humidity also showed the strongest correlation and wind speed showed more importance
than both maximum and minimum air temperature. As expected, relative humidity showed the
strongest correlation with ETo (r = -0.83) at the annual scale, followed by wind speed (r = 0.62). On
the contrary, the correlation with maximum air temperature was statistically non-significant.
The general increase observed in ETo in the Canary Islands was largely determined by changes in
the different meteorological variables (Table 7). The maximum air temperature does not show
noticeable changes, with the exception of Gran Canaria/Airport, Lanzarote and San Cristóbal
stations where significant trends were found. The regional average did not show significant
changes. On the contrary, the minimum air temperature showed an average increase of 0.12 ºC
decade$^{-1}$ in summer and 0.09 ºC decade$^{-1}$ at the annual scale between 1961 and 2013. The
significant increase recorded in summer was found in six meteorological stations, with a maximum
of 0.25º C decade$^{-1}$ in Izaña. Changes in relative humidity were also significant. There was a
significant decrease in winter, summer and annually, which represent a decline of 0.47% decade$^{-1}$,



although there were differences among stations. Sunshine duration and wind speed did not show
noticeable changes, and the unique remarkable pattern was the significant increase of the summer
sunshine duration at the regional scale (0.12 hours decade$^{-1}$) and the significant increase of wind
speed in the station of Los Rodeos in the four seasons and also annually.
With respect to the sensitivity of changes in ETo to its five driving meteorological drivers (Figure
7), substantial differences were found between variables. The differences between observed ETo
and simulated ETo with average maximum and minimum air temperature were small irrespective of
the season, indicating a low sensitivity to these two variables. In contrast, ETo was more sensitive
to setting sunshine duration and wind speed at their mean values. Thus, in the station of Los
Rodeos, the predicted magnitude of change in winter, autumn and annually was different from  the
observed magnitude of change. The highest sensitivity was, however, to relative humidity. In
general, the different meteorological stations showed an important increase in observed ETo with
respect to predicted ETo keeping relative humidity as constant. This was observed at the seasonal
and annual scales. Thus, in three meteorological stations the observed magnitude of change on
annual basis is between two and three times higher than that predicted considering relative humidity
as stationary. This pattern was also found in the regional series (Figure 8). Considering air
temperature, sunshine duration and wind speed as constant, there were no statistical differences
between the observed and predicted magnitudes of change, both seasonally and annually. On the
contrary, leaving relative humidity as constant, the magnitude of the trend was quite different to the
observations, and temporal trends would not be statistically significant. Thus, the magnitude of
change of ETo, considering relative humidity as constant, is significantly different from the the
observed magnitude of change in winter and annually.

**4. Discussion**



This work analyses the recent evolution (1961-2013) of reference evapotranspiration (ETo) in the
Canary Islands and its relationship with the evolution of its atmospheric drivers. We analysed the
time evolution of ETo in eight meteorological stations in which the necessary meteorological
variables for calculation of the ETo were available. The results showed a general increase in ETo,
although different magnitudes of change were found between the different meteorological stations.
These differences did not follow any specific geographic pattern, so they must be considered either
due to random effects and uncertainty at various levels or due to micro-geographic effects that were
not considered in this study. Nevertheless, with the exception of the observatory of San Cristóbal in
the north of Gran Canaria Island, other meteorological observatories showed positive changes in
ETo, with annual trends statistically significant in six stations. The few existing studies in
Northwest Africa (Ouysse et al., 2010; Teken and Kropp, 2012) are not comparable with our
findings, since the variables required to apply the Penman-Monteith equation were not available.
Instead, these studies relied on simplified methods that just employ air temperature records. Despite
the difference in methods, these studies also found a general increase in the ETo. The closest region
in which it is possible to make a direct comparison using the same method is the Iberian Peninsula,
where a general increase of 24.5 mm decade$^{-1}$ was found between 1961 and 2011 (Vicente-Serrano
et al., 2014a). This study also found that the variability and trends in the aerodynamic component
determined most of the observed variability and the magnitude of change of ETo in a majority of
the meteorological stations in the Iberian Peninsula. The radiative component showed much lower
temporal variability than the aerodynamic component did. Thus, more than 90% of the observed
ETo variability at the seasonal and annual scales can be associated with the variability of the
aerodynamic component. This is in agreement with the results obtained in previous studies. For
example, Wang et al. (2012) showed that recent ETo variability at the global scale was mainly
driven by the aerodynamic component. Equally, other studies in Southern Europe indicated a higher
importance of the aerodynamic component (Sanchez-Lorenzo et al., 2014; Azorin-Molina et al.,





2015). It could be argued, however, that quantification of the radiative component in our study was
based on a simplified assumption since it was calculated from sunshine duration that is mostly
determined by the cloud coverage (Hoyt, 1978). Nevertheless, it is also worth noting that global
radiation measurements, sunshine duration records contain a signal of the direct effects of aerosols
(Sanroma et al., 2010; Sanchez-Romero et al., 2014; Wild, 2015) in the Canary Islands.
Nevertheless, the Canary Islands is a region mostly free of anthropogenic aerosols given the large
frequency and intensity of trade winds (Mazorra et al., 2007), and it is not expected that the
frequency of Saharan dust events, that could affect incoming solar radiation, has noticeably changed
over the last decades (Flentje et al., 2015; Laken et al., 2015). Consequently, in the Canary Islands
we can consider high accuracy determining the radiative component using sunshine duration series.
García et al. (2014) compared the capability of sunshine duration series to reconstruct long term
radiation in the observatory of Izaña (Tenerife), showing very good temporal agreement between
sunshine duration and radiation, independently of the season of the year. In continental Spain,
Azorin-Molina et al. (2015) also found strong positive correlations between interannual variations
of solar radiation and sunshine duration in different meteorological stations. Overall, in the Canary
Islands there is a positive and significant correlation between inter-annual variations of ETo and
sunshine duration, although this correlation did not explain the observed trends of ETo in the
region.
We showed that the temporal variability of ETo is strongly controlled by the temporal variability of
relative humidity. Specifically, seasonal and annual series of ETo in the different stations showed
very strong negative and significant correlations with those of the relative humidity. Thus, the
magnitude of correlations were much higher than those obtained for other meteorological variables,
and this finding was common to the whole set of meteorological stations. This strong control of
relative humidity on the temporal variability of ETo has been already identified in some studies in





the Iberian Peninsula (Vicente-Serrano et al., 2014b; Azorin-Molina et al., 2015; Espadafor et al.,

349   2013).

Among the variables that control the aerodynamic component, wind speed and maximum air
temperature did not show significant trends at the regional scale and only few stations recorded
significant trends in these variables, either at the seasonal or the annual scales. Significant trends
were obtained for minimum air temperature, mainly in summer. Recently, Croper and Hanna (2014)
analysed long term climate trends in the Macaronesia region, and for the Canary Islands they
showed an increase in air temperature during summer for the period 1981-2010. Martín et al. (2012)
analysed air temperature changes in the Tenerife Island from 1944 to 2010 and they also showed
that night-time air temperature increased rapidly compared to daytime temperature. Nevertheless,
they found strong spatial contrasts between the high mountains, that showed a higher increase, and
the coastal areas in which the air temperature regulation of the ocean could be reducing the general
air temperature increase.
In any case, the variable that recorded more significant changes in the Canary Islands was relative
humidity, and among the different meteorological variables used to calculate ETo, relative humidity
was the main driver of the observed ETo trends. Significant negative humidity trends were recorded
in winter, summer and autumn, but also annually. Thus, simulation of ETo series considering the
different meteorological variables as constant produced few differences in relation to the observed
evolution of ETo, with the exception of the relative humidity. Leaving relative humidity as constant
for the period 1961-2013 showed no significant ETo changes at seasonal and annual scales and also
statistically significant differences with changes obtained from observations. In continental Spain,
Vicente-Serrano et al. (2014b) showed a general decrease of relative humidity from the decade of
1960, mainly associated with a general decrease of the moisture transport to the Iberian Peninsula as
well as a certain precipitation decrease. Similarly, Espadafor et al. (2011) and Vicente-Serrano et al.
(2014b) showed that the strong increase in ETo in the last decades is associated with the relative





humidity decrease due to air temperature rise. In the Canary Islands, no precipitation changes have
been identified during the analyzed period (Sánchez-Benitez et al., 2015). Therefore a lower
moisture supply from the humidity sources to the islands should explain the observed pattern
toward a relative humidity decrease. Sherwood and Fu (2014) suggested that differences in the air
temperature increase between oceanic and continental areas could increase land aridity, as a
consequence of the sub-saturation conditions of the oceanic air masses that come to the land areas,
given higher warming rates in maritime regions in comparison to continental areas. The results of
this study confirm this pattern in the Canary Islands, since this region should not be constrained by
constant moisture supply from the surrounding warm Atlantic Ocean. Overall, Willett et al. (2014)
recently found a general decrease in relative humidity at the global scale, including several islands
and coastal regions in which the moisture supply was expected to be unlimited. This finding
suggests that contrasted mean air temperature and trends between land and ocean areas could also
play an important role in explaining this phenomenon, even at local scales.

**5. Conclusions**
We found that the reference evapotranspiration ETo increased by 18.2 mm decade$^{-1}$ -on average-
between 1961 and 2013 over the Canary Islands, with the highest increase recorded during summer.
Although there were noticeable spatial differences, this increase was mainly driven by changes in
the aerodynamic component, caused by a statistically significant reduction of the relative humidity.
This study provides an outstanding example of how climate change and interactions between
different meteorological variables drive an increase of the ETo event in a subtropical North Atlantic
Islands. Given the general aridity conditions in most of the Canary Islands and the scarcity of water
resources, the observed trend could have negative consequences in a number of water-depending
sectors if it continues in the future.





**Acknowledgements**
The authors thank Spanish Meterological Agency (AEMET) for providing the climate data used in
this study. This work was supported by the research project CGL2014-52135-C03-01, *Red de*
*variabilidad y cambio climático* RECLIM (CGL2014-517221-REDT) financed by the Spanish
Commission of Science and Technology and FEDER and "LIFE12 ENV/ES/000536-Demonstration
and validation of innovative methodology for regional climate change adaptation in the
Mediterranean area (LIFE MEDACC)" financed by the LIFE programme of the European
Commission. Cesar Azorin-Molina (JCI-2011-10263), Arturo Sanchez-Lorenzo (JCI-2012-12508)
and Marina Peña-Gallardo were granted by the Spanish Ministry of Economy and Competitiveness;
Natalia Martin-Hernandez was supported by a doctoral grant by the Aragón Regional Government;
and Miquel Tomas-Burguera was supported by a doctoral grant by the Ministerio de Educación,
Cultura y Deporte.

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

numberD12110).



Table 1: Seasonal and annual averages (mm) and coefficients of variation of ETo in the eight
meteorological stations, averaged over the period 1961-2013.

| | Sta. Cruz de Tenerife | Gran Canaria/Airp. | Los Rodeos | Izaña | Fuerteventura | La Palma | Lanzarote | San Cristóbal | Regional Mean |
|---|---|---|---|---|---|---|---|---|---|
| | | | | | **Average** | | | | |
| **Winter** | 222.0 | 181.5 | 297.5 | 250.2 | 298.1 | 251.6 | 294.5 | 217.7 | 251.6 |
| **Spring** | 390.1 | 302.2 | 468.8 | 414.1 | 460.8 | 361.5 | 468.7 | 342.3 | 401.1 |
| **Summer** | 512.7 | 415.5 | 612.9 | 663.8 | 560.2 | 438.7 | 586.1 | 383.0 | 521.6 |
| **Autumn** | 311.8 | 273.9 | 401.8 | 364.5 | 384.6 | 316.4 | 393.8 | 278.8 | 340.7 |
| **Annual** | 1435.5 | 1175.0 | 1784.4 | 1692.6 | 1702.0 | 1372.7 | 1741.0 | 1219.4 | 1515.3 |
| | | | | | **Coefficient of variation** | | | | |
| **Winter** | 0.05 | 0.11 | 0.12 | 0.18 | 0.10 | 0.11 | 0.09 | 0.11 | 0.06 |
| **Spring** | 0.04 | 0.10 | 0.07 | 0.12 | 0.08 | 0.10 | 0.06 | 0.08 | 0.05 |
| **Summer** | 0.03 | 0.12 | 0.07 | 0.07 | 0.07 | 0.08 | 0.07 | 0.07 | 0.04 |
| **Autumn** | 0.03 | 0.10 | 0.10 | 0.10 | 0.07 | 0.11 | 0.07 | 0.08 | 0.05 |
| **Annual** | 0.02 | 0.07 | 0.06 | 0.07 | 0.07 | 0.08 | 0.06 | 0.05 | 0.04 |





Table 2: Seasonal and annual averages (mm) and coefficients of variation of aerodynamic and
radiative components of ETo in the eight meteorological stations.In bold the values greater than
50% of the total ETo of the station

| | Sta. Cruz de Tenerife | Gran Canaria/Airp. | Los Rodeos | Izaña | Fuerteventura | La Palma | Lanzarote | San Cristóbal | Mean |
|---|---|---|---|---|---|---|---|---|---|
| | | | | | **Aerodynamic** | | | | |
| | | | | | Average | | | | |
| Winter | 101.6 | **98.8** | **198.8** | **198.8** | **195.9** | **153.2** | **190.4** | 108.1 | **155.7** |
| Spring | 130.5 | 137.5 | **287.2** | **271.0** | **251.1** | 174.3 | **262.0** | 134.7 | **206.0** |
| Summer | 146.2 | 195.6 | **394.7** | **424.7** | **288.5** | 201.7 | **328.1** | 143.1 | **265.3** |
| Autumn | 109.3 | 133.4 | **249.1** | **263.6** | **211.7** | 157.6 | **225.9** | 102.0 | **181.6** |
| Annual | 487.5 | 568.0 | **1134.4** | **1158.6** | **945.8** | 690.7 | **1004.4** | 485.5 | **809.4** |
| | | | | | Coefficient of variation | | | | |
| Winter | 0.12 | 0.19 | 0.22 | 0.23 | 0.18 | 0.19 | 0.16 | 0.27 | 0.11 |
| Spring | 0.11 | 0.18 | 0.15 | 0.17 | 0.16 | 0.20 | 0.12 | 0.26 | 0.09 |
| Summer | 0.13 | 0.24 | 0.12 | 0.14 | 0.15 | 0.18 | 0.12 | 0.20 | 0.08 |
| Autumn | 0.13 | 0.21 | 0.20 | 0.14 | 0.14 | 0.20 | 0.15 | 0.25 | 0.10 |
| Annual | 0.09 | 0.16 | 0.13 | 0.12 | 0.14 | 0.16 | 0.11 | 0.17 | 0.07 |
| | Sta. Cruz de Tenerife | Gran Canaria/Airp. | Los Rodeos | Izaña | Fuerteventura | La Palma | Lanzarote | San Cristóbal | Average |
| | | | | | **Radiative** | | | | |
| | | | | | Average | | | | |
| Winter | **120.4** | 82.7 | 98.6 | 51.4 | 102.2 | 98.4 | 104.1 | **109.6** | 95.9 |
| Spring | **259.7** | **164.7** | 181.5 | 143.1 | 209.7 | **187.2** | 206.7 | **207.6** | 195.0 |
| Summer | **366.5** | **220.0** | 218.3 | 239.1 | 271.7 | **237.0** | 258.0 | **240.0** | 256.3 |
| Autumn | **202.4** | **140.5** | 152.8 | 100.9 | 172.9 | **158.8** | 167.9 | **176.8** | 159.1 |
| Annual | **948.1** | **607.0** | 650.0 | 534.0 | 756.3 | 682.0 | 736.7 | **734.0** | 706.0 |
| | | | | | Coefficient of variation | | | | |
| Winter | 0.05 | 0.08 | 0.10 | 0.12 | 0.08 | 0.08 | 0.09 | 0.08 | 0.06 |
| Spring | 0.06 | 0.07 | 0.08 | 0.09 | 0.06 | 0.07 | 0.06 | 0.08 | 0.05 |
| Summer | 0.04 | 0.06 | 0.07 | 0.08 | 0.05 | 0.09 | 0.06 | 0.10 | 0.04 |
| Autumn | 0.05 | 0.05 | 0.08 | 0.07 | 0.05 | 0.06 | 0.06 | 0.06 | 0.04 |
| Annual | 0.03 | 0.04 | 0.07 | 0.06 | 0.04 | 0.05 | 0.04 | 0.06 | 0.03 |







Table 3: Magnitude of change (mm. decade$^{-1}$) of ETo in each meteorological station and the average of the
eight stations over the period 1961-2013. Statistically significant at the 95% confidence level are given in
bold. Numbers between brackets refer to the magnitudes of change for the periods 1933-2013 for Izaña and

1943-2013 for Santa Cruz de Tenerife.

|  | Sta. Cruz de Tenerife | Gran Canaria/Airp. | Los Rodeos | Izaña | Fuerteventura | La Palma | Lanzarote | San Cristóbal | Mean |
|---|---|---|---|---|---|---|---|---|---|
| Winter | **2.7**(0.31) | 1.7 | **11.3** | 4.8 (-0.42) | 3.2 | **9.1** | **7.1** | **-5.1** | **4.3** |
| Spring | 0.1 (-0.55) | **7.7** | **7.1** | -0.1 (-1.27) | 3.9 | **7.2** | 4.0 | **-5.8** | 3.0 |
| Summer | 1.1 (-1.36) | **16.0** | **7.6** | 6.0 (-0.64) | 0.0 | **7.7** | **10.1** | 5.0 | 6.7 |
| Autumn | **2.0**(0.62) | 3.6 | **11.2** | 3.7 (0.30) | -0.2 | **9.9** | 4.8 | **-5.0** | 3.8 |
| Annual | **7.3**(-1.95) | **28.4** | **34.8** | 14.9 (-0.67) | 9.2 | **29.8** | **29.7** | -8.4 | **18.2** |




Table 4: Magnitude of change (mm. decade$^{-1}$) of both aerodynamic and radiative components of ETo in each
meteorological station and the average of the eight stations over the period 1961-2013. Statistically
significant at the 95% confidence level are given in bold. Numbers between brackets refer to the magnitudes
of change for the periods 1933-2013 for Izaña and 1943-2013 for Santa Cruz de Tenerife.



| | Sta. Cruz de Tenerife | Gran Canaria/Airp. | Los Rodeos | Izaña | Fuerteventura | La Palma | Lanzarote | San Cristóbal | **Mean** |
|---|---|---|---|---|---|---|---|---|---|
| | | | | **Aerodynamic** | | | | | |
| Winter | **3.7** (0.09) | 2.9 | **14.8** | 5.1 (-0.96) | 4.6 | **10.1** | **9.1** | **-5.8** | **5.5** |
| Spring | -1.3 (-1.84) | **7.8** | **8.9** | 0.1 (-3.39) | 2.4 | 3.3 | 2.7 | **-11.8** | 1.5 |
| Summer | 0.1 (-2.95) | **16.8** | **9.9** | 6.7 (-3.38) | -1.1 | 2.5 | **8.1** | -1.5 | **5.2** |
| Autumn | 2.1 (-0.51) | **5.2** | **14.5** | 3.7 (-1.03) | -1.1 | **7.9** | 4.6 | -3.8 | **4.1** |
| Annual | 4.7 (-6.25) | **31.2** | **44.6** | 15.6 (-6.93) | 6.5 | 19.8 | **28.0** | **-21.2** | **16.2** |
| | | | | **Radiative** | | | | | |
| Winter | -1.0 (0.22) | **-1.2** | **-3.5** | -0.4 (0.51) | -1.4 | -1.0 | -2.0 | 0.8 | **-1.2** |
| Spring | 1.4 (1.28) | -0.1 | -1.8 | -0.3 (2.12) | 1.4 | **3.9** | 1.3 | **6.1** | 1.5 |
| Summer | 1.0 (1.58) | -0.8 | -2.3 | -0.7 (2.74) | 1.1 | **5.1** | 2.0 | **6.5** | 1.5 |
| Autumn | 0.0 (1.13) | **-1.6** | **-3.3** | 0.1 (1.34) | 0.9 | **2.0** | 0.2 | -1.2 | -0.4 |
| Annual | 2.7 (4.29) | -2.8 | **-9.9** | -0.7 (6.25) | 2.7 | **10.0** | 1.7 | **12.7** | 2.0 |



Table 5. Seasonal and annual Pearson's coefficients between the evolution of ETo and the evolution of
aerodynamic and radiative components in the eight meteorological stations and the average. Statistically
significant at the 95% confidence level are given in bold




|  | Sta. Cruz de Tenerife | Gran Canaria/Airp. | Los Rodeos | Izaña | Fuerteventura | La Palma | Lanzarote | San Cristóbal | **Mean** |
|---|---|---|---|---|---|---|---|---|---|
| Aerodynamic | | | | | | | | | |
| Winter | **0.88** | **0.95** | **0.99** | **0.99** | **0.98** | **0.97** | **0.97** | **0.96** | **0.93** |
| Spring | **0.65** | **0.93** | **0.95** | **0.96** | **0.95** | **0.93** | **0.93** | **0.88** | **0.87** |
| Summer | **0.74** | **0.96** | **0.96** | **0.97** | **0.94** | **0.84** | **0.94** | **0.63** | **0.85** |
| Autumn | **0.75** | **0.96** | **0.98** | **0.98** | **0.96** | **0.96** | **0.97** | **0.90** | **0.95** |
| Annual | **0.78** | **0.97** | **0.98** | **0.97** | **0.97** | **0.95** | **0.96** | **0.88** | **0.95** |
| Radiative | | | | | | | | | |
| Winter | 0.05 | **0.37** | **-0.75** | **0.18** | **-0.62** | -0.22 | **-0.44** | **-0.46** | -0.02 |
| Spring | **0.38** | **0.52** | **-0.51** | **0.36** | -0.25 | 0.14 | 0.07 | -0.17 | **0.28** |
| Summer | 0.05 | **0.28** | **-0.37** | **-0.62** | -0.12 | 0.23 | 0.08 | **0.41** | **0.29** |
| Autumn | 0.14 | 0.09 | **-0.67** | -0.01 | -0.23 | **0.43** | **-0.45** | -0.05 | 0.05 |
| Annual | -0.05 | -0.20 | **-0.73** | **-0.36** | **-0.46** | 0.04 | **-0.28** | **-0.29** | -0.15 |





Table 6. Seasonal and annual Pearson's coefficients between the time series of ETo and the different
meteorological variables in the eight meteorological stations, calculated for the period 1961-2013.
Statistically significant at the 95% confidence level are given in bold




| | Sta. Cruz de Tenerife | Gran Canaria/Airp. | Los Rodeos | Izaña | Fuerteventura | La Palma | Lanzarote | San Cristóbal |
|---|---|---|---|---|---|---|---|---|
| **Maximum air temperature** | | | | | | | | |
| Winter | **0.32** | **0.51** | -0.12 | **0.89** | -0.23 | -0.01 | -0.23 | 0.26 |
| Spring | **0.46** | **0.69** | 0.02 | **0.90** | 0.18 | 0.01 | **0.62** | **0.42** |
| Summer | **0.48** | **0.80** | 0.10 | 0.18 | **0.33** | 0.27 | **0.51** | **0.44** |
| Autumn | 0.18 | **0.64** | 0.04 | **0.71** | **0.29** | 0.12 | 0.09 | **0.43** |
| Annual | 0.17 | **0.41** | -0.11 | **0.64** | 0.01 | -0.03 | 0.16 | **0.46** |
| **Minimum air temperature** | | | | | | | | |
| Winter | 0.15 | **0.50** | 0.13 | **0.83** | -0.24 | 0.17 | -0.13 | 0.01 |
| Spring | 0.24 | **0.53** | 0.19 | **0.83** | 0.12 | 0.19 | **0.49** | 0.10 |
| Summer | 0.24 | **0.55** | 0.11 | 0.23 | 0.16 | **0.33** | **0.55** | 0.17 |
| Autumn | 0.21 | **0.56** | **0.36** | **0.63** | 0.20 | **0.32** | **0.26** | 0.21 |
| Annual | 0.04 | **0.47** | 0.13 | **0.54** | -0.11 | **0.30** | 0.27 | -0.07 |
| **Relative humidity** | | | | | | | | |
| Winter | **-0.52** | **-0.91** | **-0.57** | **-0.83** | **-0.92** | **-0.92** | **-0.89** | **-0.72** |
| Spring | **-0.34** | **-0.89** | **-0.70** | **-0.90** | **-0.89** | **-0.90** | **-0.77** | **-0.82** |
| Summer | **-0.35** | **-0.93** | **-0.83** | **-0.46** | **-0.90** | **-0.89** | **-0.80** | **-0.61** |
| Autumn | **-0.30** | **-0.94** | **-0.55** | **-0.74** | **-0.90** | **-0.91** | **-0.78** | **-0.76** |
| Annual | -0.18 | **-0.93** | **-0.62** | **-0.59** | **-0.93** | **-0.94** | **-0.85** | **-0.86** |
| **Sunshine duration** | | | | | | | | |
| Winter | **0.48** | **0.48** | 0.16 | **0.63** | 0.01 | **0.33** | 0.18 | 0.06 |
| Spring | **0.72** | **0.71** | 0.08 | **0.70** | 0.27 | **0.50** | 0.25 | 0.21 |
| Summer | **0.45** | **0.62** | 0.20 | 0.18 | **0.32** | **0.41** | **0.35** | **0.61** |
| Autumn | **0.47** | **0.38** | 0.20 | **0.53** | 0.14 | **0.69** | 0.16 | **0.34** |
| Annual | **0.40** | **0.30** | -0.01 | **0.40** | 0.15 | **0.48** | 0.08 | -0.09 |
| **Wind speed** | | | | | | | | |
| Winter | **0.61** | -0.01 | **0.84** | 0.29 | **0.54** | 0.29 | **0.35** | **0.62** |
| Spring | **0.47** | 0.18 | **0.62** | 0.33 | **0.52** | 0.22 | 0.24 | **0.44** |
| Summer | **0.65** | 0.37 | 0.48 | 0.77 | **0.39** | -0.01 | **0.33** | 0.26 |
| Autumn | **0.62** | 0.22 | **0.78** | 0.48 | **0.31** | 0.27 | **0.62** | 0.48 |
| Annual | **0.73** | **0.47** | **0.72** | **0.69** | **0.50** | 0.25 | **0.34** | **0.38** |



Table 7. Magnitude of change (ºC, %, hours and ms$^{-1}$ decade$^{-1}$) of the different meteorological variables over the period 1961-2013. In bold statistically significant trends at the 95%.

| | Sta. Cruz de Tenerife | Gran Canaria/Airp. | Los Rodeos | Izaña | Fuerteventura | La Palma | Lanzarote | San Cristóbal | Mean |
|---|---|---|---|---|---|---|---|---|---|
| **Maximum air temperature** | | | | | | | | | |
| Winter | -0.06 | -0.09 | -0.05 | -0.01 | -0.08 | -0.08 | **-0.18** | **-0.18** | -0.09 |
| Spring | -0.08 | 0.03 | -0.02 | -0.12 | -0.02 | -0.02 | 0.08 | 0.14 | 0.00 |
| Summer | -0.06 | **0.20** | 0.00 | -0.07 | 0.00 | 0.00 | 0.07 | 0.12 | 0.04 |
| Autumn | -0.06 | -0.08 | -0.08 | -0.04 | -0.10 | -0.06 | -0.11 | **-0.17** | -0.09 |
| Annual | -0.05 | 0.03 | -0.01 | -0.05 | -0.03 | -0.02 | -0.01 | 0.00 | -0.02 |
| **Minimum air temperature** | | | | | | | | | |
| Winter | -0.02 | -0.01 | 0.02 | 0.16 | -0.02 | 0.02 | -0.02 | 0.14 | 0.03 |
| Spring | 0.02 | 0.03 | 0.03 | 0.18 | 0.04 | 0.04 | 0.05 | 0.09 | 0.06 |
| Summer | 0.08 | **0.12** | **0.10** | **0.25** | **0.11** | 0.07 | **0.10** | **0.13** | **0.12** |
| Autumn | 0.07 | 0.01 | **0.09** | **0.19** | 0.05 | 0.09 | 0.09 | 0.08 | **0.09** |
| Annual | 0.05 | 0.05 | 0.08 | **0.20** | 0.06 | 0.07 | 0.08 | **0.12** | **0.09** |
| **Relative humidity** | | | | | | | | | |
| Winter | **-0.51** | -0.51 | -0.22 | -1.11 | -0.81 | **-1.53** | **-1.56** | -0.18 | **-0.80** |
| Spring | 0.18 | **-1.06** | -0.22 | 0.20 | -0.76 | -0.96 | **-0.88** | **0.90** | -0.33 |
| Summer | 0.39 | **-1.58** | -0.16 | -0.91 | -0.06 | -0.72 | **-0.99** | 0.45 | **-0.45** |
| Autumn | 0.02 | -0.72 | 0.01 | -0.26 | -0.29 | **-1.65** | **-0.99** | 0.31 | **-0.45** |
| Annual | 0.02 | **-0.89** | -0.03 | -0.52 | -0.49 | **-1.05** | **-1.11** | 0.32 | **-0.47** |
| **Sunshine duration** | | | | | | | | | |
| Winter | 0.02 | **-0.10** | -0.04 | 0.02 | -0.12 | 0.08 | -0.05 | **-0.11** | -0.04 |
| Spring | 0.08 | 0.11 | 0.08 | 0.06 | 0.03 | **0.22** | -0.06 | 0.05 | 0.07 |
| Summer | 0.06 | **0.15** | 0.05 | -0.03 | 0.00 | **0.25** | 0.09 | **0.35** | **0.12** |
| Autumn | 0.03 | -0.04 | 0.03 | 0.08 | 0.00 | 0.19 | 0.03 | **-0.16** | 0.02 |
| Annual | 0.06 | 0.03 | 0.03 | 0.04 | -0.01 | **0.18** | 0.02 | 0.04 | 0.05 |
| **Wind speed** | | | | | | | | | |
| Winter | **0.04** | 0.04 | **0.33** | 0.01 | 0.00 | **0.07** | 0.02 | **-0.18** | 0.04 |
| Spring | -0.01 | 0.08 | **0.19** | 0.07 | -0.08 | -0.08 | **-0.13** | **-0.24** | -0.03 |
| Summer | 0.02 | **0.21** | 0.24 | -0.01 | -0.05 | -0.11 | -0.06 | 0.01 | 0.03 |
| Autumn | 0.03 | 0.07 | **0.33** | 0.03 | -0.07 | -0.05 | -0.04 | -0.06 | 0.03 |
| Annual | 0.02 | **0.10** | **0.27** | 0.02 | -0.04 | -0.04 | -0.04 | **-0.12** | 0.02 |





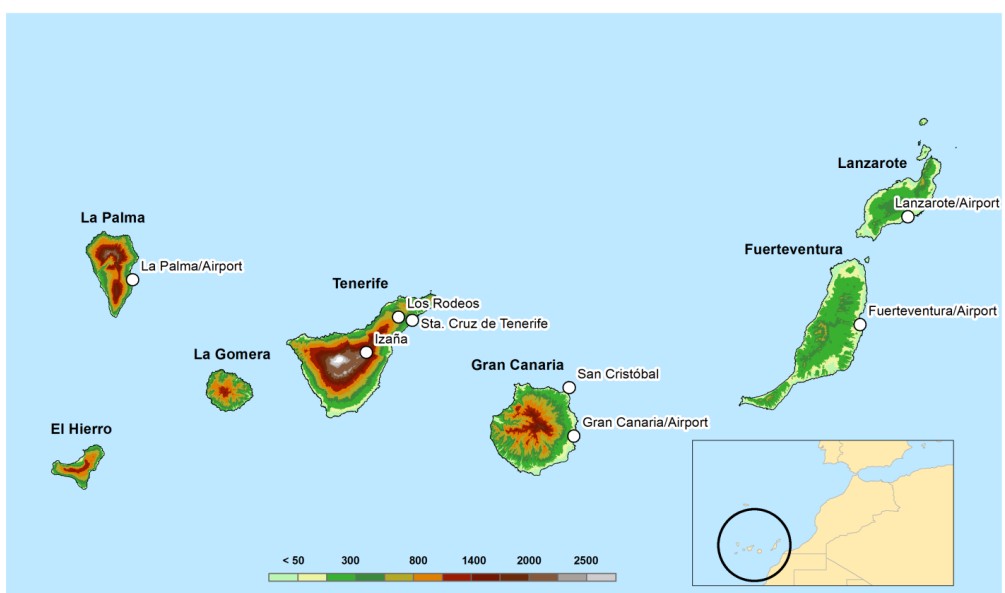


Figure 1: Location and relief of the Canary Islands and meteorological stations used in the study. Altitude is given in meters.







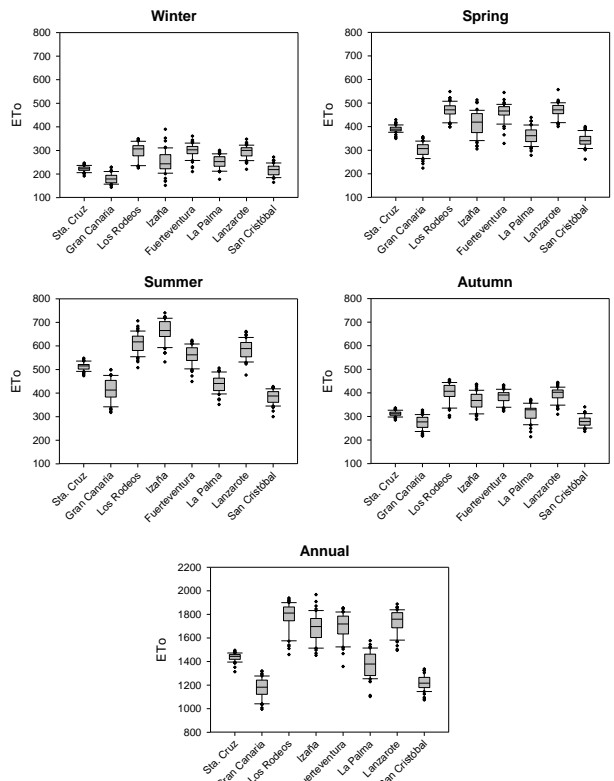


Figure 2: Box-plot with the annual and seasonal ETo values in the eight meteorological stations
used in this study. The vertical lines of each plotted boxplot illustrate the 10th, 25th, 75th and 90th
quantiles, respectively. The interquartile spread is represented by the range between the 25th and
75th quantiles. The dots show the highest and lowest values.






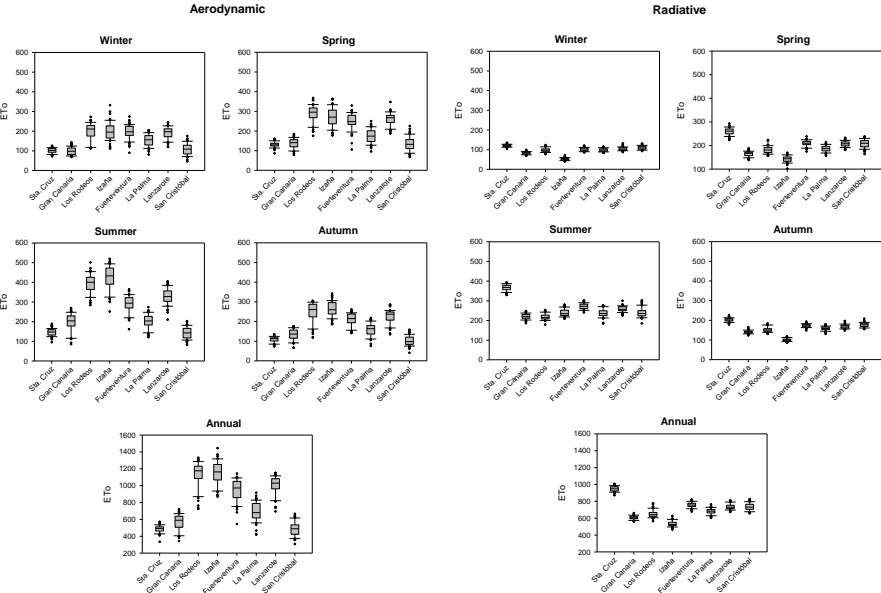


Figure 3: Box-plot with the annual and seasonal aerodynamic and radiative components of ETo in
the eight meteorological stations used in this study








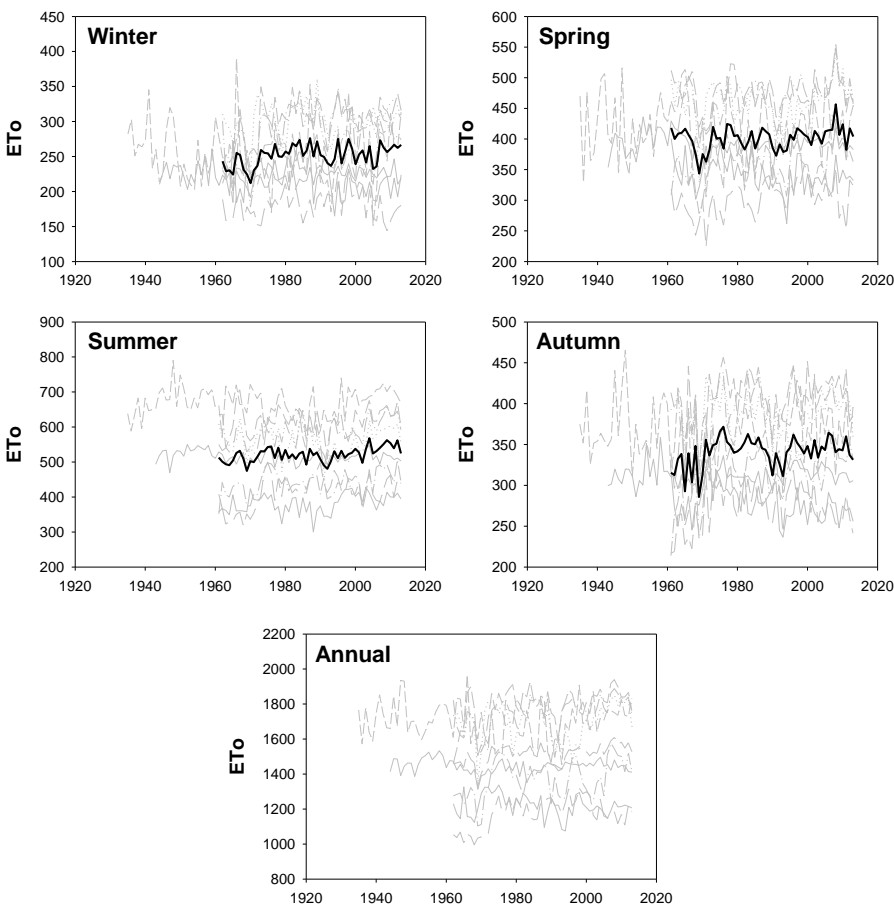


Figure 4: Evolution of seasonal and annual ETo in the eight meteorological stations (grey lines) and
the average of the eight stations (black lines) from 1961 to 2013.





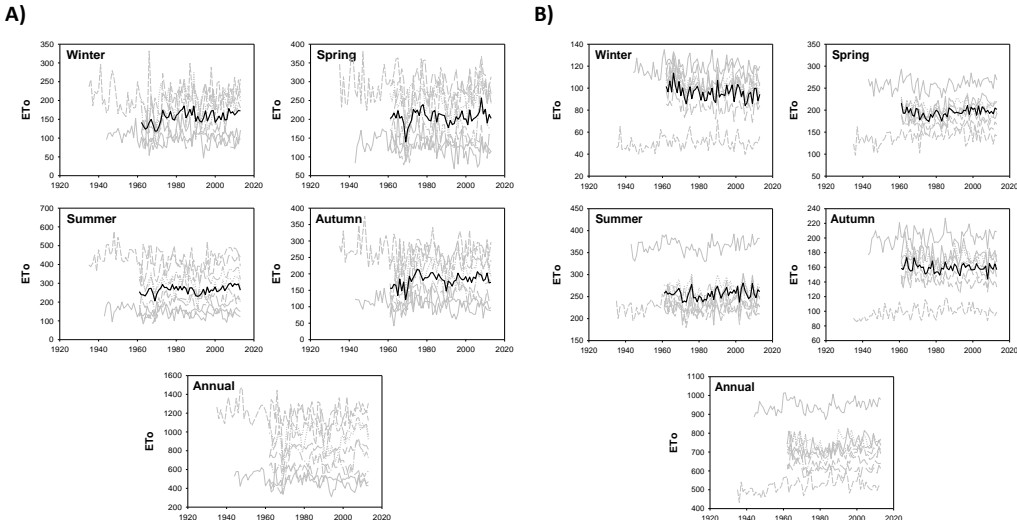


Figure 5: Evolution of seasonal and annual aerodynamic (A) and radiative (B) components of the
ETo in the eight meteorological stations (grey lines) and the average of the eight stations (black
lines) from 1961 to 2013






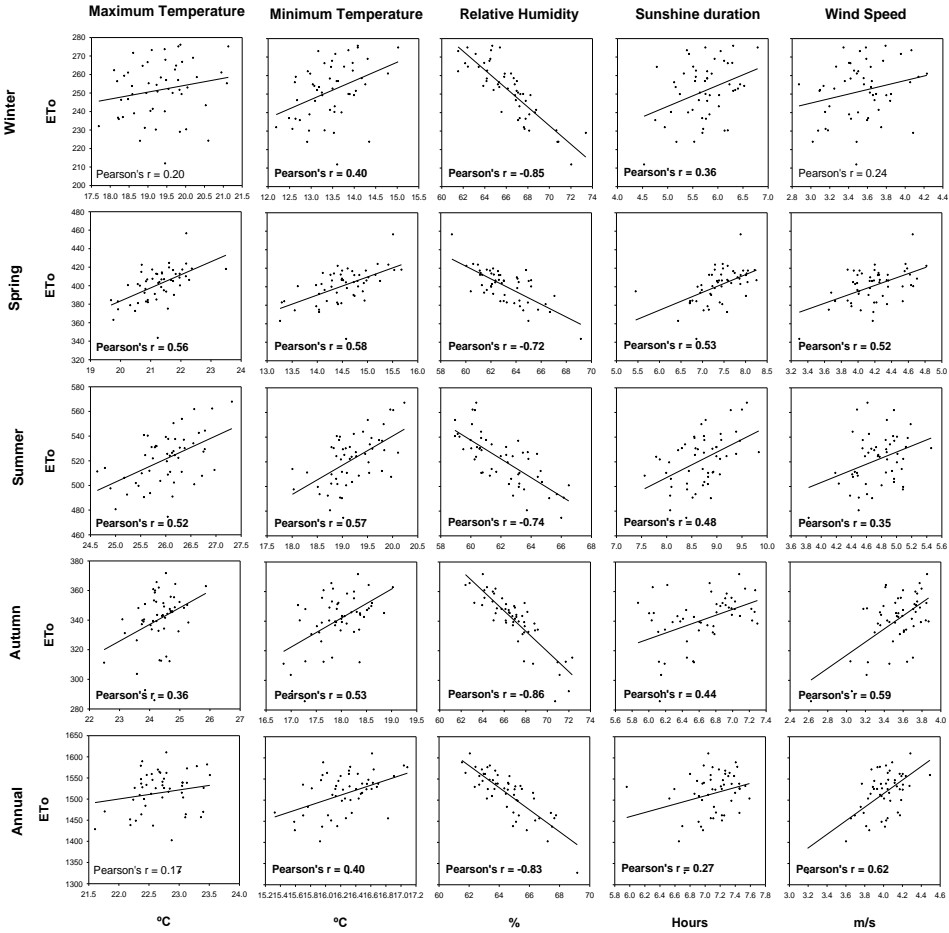


Figure 6. Relationship between the regional annual and seasonal ETo and the regional series of the
different meteorological variables. Pearson's coefficients are included in each plot. In bold the
coefficients statistically significant at the 0.95 confidence level








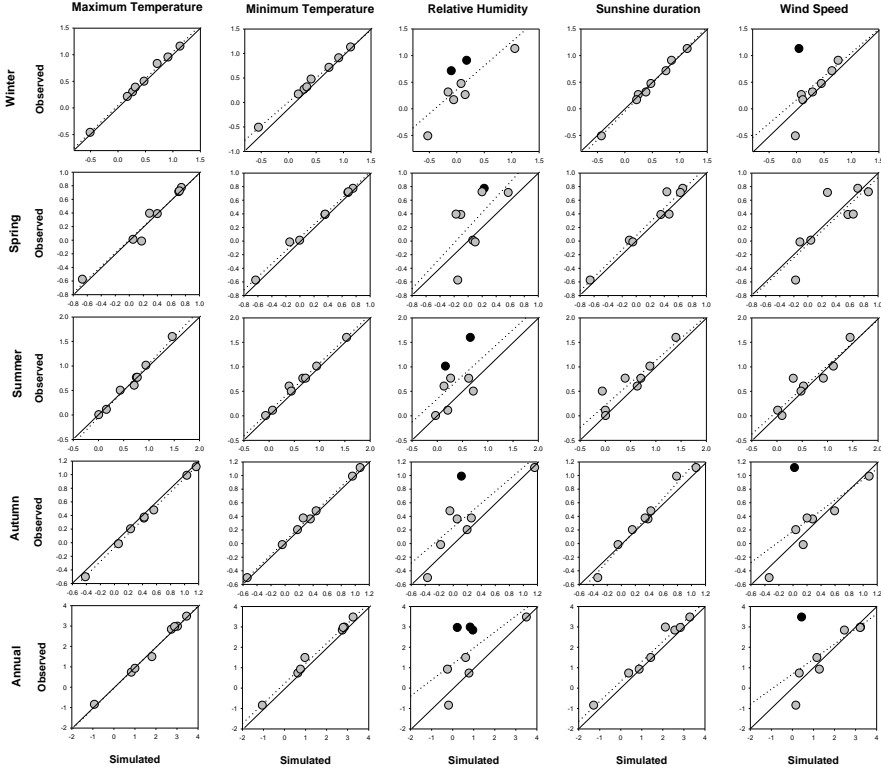



Figure 7: Relationship between the seasonal and annual observed magnitude of change of ETo
(mm. year-1) in each meteorological station and the simulated magnitude of change maintaining
each meteorological variable as constant. Black dots indicate significant differences in the trends.





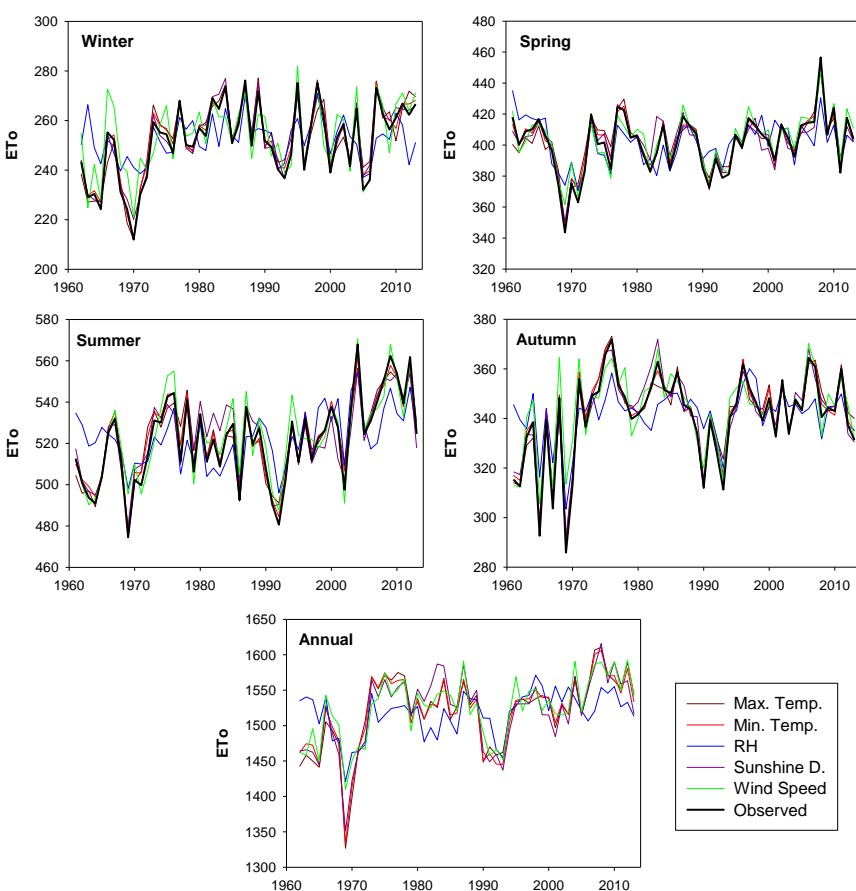

Figure 8: Seasonal and annual evolution of the observed regional ETo compared to the  simulated
ETo considering no temporal changes in each one of the meteorological variables from 1961 to
669    2013.