# Peer review of "Recent changes and drivers of the atmospheric evaporative demand in the Canary Islands"

_Hydrology and Earth System Sciences, 2016_

## Referee Comment (RC1) · Anonymous Referee #1 · 16 Apr 2016

The manuscript presents a trend analysis of the FAO-56 reference evaporation using meteorological data from 8 sites at the Canary Islands. Interestingly the results show a remarkable heterogeneity in both the drivers of ET0 and its trends, which I did not expect due to the maritime climate. The most consistent effect is the decrease in relative humidity at most sites and thus an increase of the aerodynamic component of ET0. Generally the paper is well written, the data analysis is comprehensive and well designed. The topic of observed ET0 changes and choice of the sites are relevant and well suited for publication in HESS. Although I have a some remarks I am positive that the authors can implement these and recommend minor revisions.

Comments and remarks

The trends in ET0 seem to be significant because of the low values in the beginning of

the chosen period. The results of the two sites with longer coverage show no significance. Thus the trend seems to be rather an effect of decadal scale variability. Please indicate this within the discussion of the results.

Abstract: L16 ET0 is not explained, please state here in the abstract that you estimate AED by the FAO-56 reference evaporation equation

L17-18 The sentence "The radiative component … did" can be removed because this is again stated in the next sentence. Also explain the meaning of the two components.

Introduction: The main research hypotheses should be clearly formulated

Section 2.1 L107-116: The homogenisation alters the original data and can effect the detection of trends. To achieve reproducibility of the results I recommend to provide an overview about data gaps, breakpoints and corrections which should be added as supplement.

Section 2.3 L160-171 I do like the simple yet illustrate way to determine the effects of single variables on the detected trend. By design this is done as a local sensitivity analysis where one variable is changed holding the others fixed. However, it is not a global sensitivity analysis and co-variation of the forcing variables is neglected. Especially for the meteorological variables used here, I suspect that the variables and eventually their trends in time do co-vary - e.g. temperature and relative humidity. Did you consider such effects and are they important to understand the long-term variability?

L195 … the aerodynamic component (Eq. 3).

Discussion: L303- 305 differences in ET0 trends across sites … "must be considered either due to random effects and uncertainty at various levels or due to micro-geographic effects … " I think the differences of the trends and the different strength of the aerodynamic / radiative components at the sites deserve more attention in the discussion. The results are presented already in a detailed manner and these aspects should be discussed. That means is there a link of the different strength of certain

forcing variables and the magnitude of ET0 trend at a given site.

L514: in preparation

Tables: Please add a Site Metadata Table with site name, WMO-ID, LAT, LON, Height, gaps, relocation, corrections, available variables

Figure 1: Please add LAT - LON coordinates as a grid and a scale for the distance

Figure 4 and 5: The grey lines are not very informative. Please adapt the figures, using different colors or line types for the sites. It might be also useful to demean the time series for the display. In the annual panel the bold line is missing.

Fig. 5 the labels are too small to be readable

---

## Referee Comment (RC2) · Anonymous Referee #2 · 29 Apr 2016

The manuscript deals with an analysis of the atmospheric evaporative demand (AED) over the Canarian Island for the period 1961-2013. Basis are meteorological data (monthly, p4l96) from 8 stations which are used as inputs for the FAO-56 Penman-Monteith equation to derive monthly AED. While the paper is generally well written, I feel there are a number of conceptuals issues that need to be resolved and addressed before a possible acceptance.

- As the FAO-56 is a non-linear equation that has been developed for daily inputs, how do authors justify the application of monthly average input values?

- As some of the input variables (Rn) have to be estimated from other parameters, some of the discussion about these relationships (p.14) need to be provided earlier in the text.

- While in general there are many graphical illusitraion for plenty of aspects, I actually miss graphs with the temporal dynamics and developments of input variables into the FAO-56 equation. Where can I see the trend for Rn, T, wind speed, rH? This would be important as they are the controlling variable in the equation.

- Why are authors relating calculated ET0 with variables that have been used to calculate ET0 before (or used to derive inputs from where ET0 is calculated) - see for example Fig. 6. Why don't authors simply calculate the sensitivities (partial derivatives) of FAO-56 with respect to the driving variables. I simply did that and only from using a temperature increase of 0.6 °C (keeping specific water content constant) and some realistic Rn, T , ra, rs – values (I used the original PM formula) I could derive the changes in ET0 stated by the authors. I feel a sensitivity study in this way including trend analysis of the inputs would be more compact and informative for the readers.

- Authors state they applied the Mann-Kandell – did they check and correct for auto-correlation?

Overall, I feel there is still a large potential to improve the overall structure/concept of the manuscript as outlined above. As a result I suggest major revisions to the manuscript before publication.
* * *

---

## Author Comment (AC1) · 25 May 2016

Interactive comment on "Recent changes and drivers of the atmospheric evaporative demand in the Canary Islands" by S. M. Vicente-Serrano et al. Anonymous Referee #1

The manuscript presents a trend analysis of the FAO-56 reference evaporation using meteorological data from 8 sites at the Canary Islands. Interestingly the results show a remarkable heterogeneity in both the drivers of ET0 and its trends, which I did not expect due to the maritime climate. The most consistent effect is the decrease in relative humidity at most sites and thus an increase of the aerodynamic component of ET0. Generally the paper is well written, the data analysis is comprehensive and well designed. The topic of observed ET0 changes and choice of the sites are relevant and

well suited for publication in HESS. Although I have a some remarks I am positive that the authors can implement these and recommend minor revisions.

We would like to thank the Reviewer#1 for his/her positive assessment of our manuscript. We are also very grateful with the detailed revision of the manuscript and the constructive comments raised with the purpose of improving our research article. They have been discussed below and implemented in this revised version.

Comments and remarks The trends in ET0 seem to be significant because of the low values in the beginning of the chosen period. The results of the two sites with longer coverage show no significance. Thus the trend seems to be rather an effect of decadal scale variability. Please indicate this within the discussion of the results.

We thank for this point and fully agree with this suggestion. This has been included in the discussion section of the revised manuscript:

"In any case, we must also stress that trends in ETo at the regional scale are mostly significant because of the low values in the beginning of the study period starting in the 1960s. Thus, the results of the two sites with longer temporal coverage (i.e., Izaña and Santa Cruz de Tenerife) do not show significant trends. This makes necessary to consider these trends with caution since they could be driven by variability processes at the decadal scale."

Abstract: L16 ET0 is not explained, please state here in the abstract that you estimate AED by the FAO-56 reference evaporation equation

The Reviewer#1 is right and we have included this information in the abstract:

"Overall, the annual ETo, which was estimated by means of the FAO-56 Penman-Monteith equation, increased significantly by. . ."

L17-18 The sentence "The radiative component . . . did" can be removed because this is again stated in the next sentence. Also explain the meaning of the two components.

The sentence has been removed as suggested and we have explained the meaning of aerodynamic and radiative components as follows:

"In this study we analysed the contribution of (i) the aerodynamic (related to the water vapour that a parcel of air can store) and (ii) radiative (related to the available energy to evaporate a quantity of water) componets to the decadal variability and trends of ETo."

Introduction: The main research hypotheses should be clearly formulated

Thanks for this suggestion. The main hypothesis of our study has been included in the introduction section:

"The main hypothesis of the study is that in opposition to other continental temperate regions of the North Hemisphere, the warm and humid climate of the subtropical Canary Islands provides the water supply to the atmosphere needed to maintain the AED constant under the current global warming scenarios; consequently, only wind speed and solar radiation could affect the observed decadal variability and trends of the AED."

Section 2.1 L107-116: The homogenisation alters the original data and can affect the detection of trends. To achieve reproducibility of the results I recommend to provide an overview about data gaps, breakpoints and corrections which should be added as supplement.

The Reviewer#1 is right when stating that homogenisation alters the original data, but this "alteration" is really necessary in any study that focuses on climate variability and trends using observed datasets. On the contrary, there is a substantial risk on the robustness of the obtained climate trends based on inhomogeneous or not quality-controlled and tested series. For this reason, independently of the alteration of the series after the homogenisation tests, the application of homogenisation methods is strictly mandatory in any climate study aimed at retrieving long-term trends.

Moreover, we completely agree with the Reviewer#1 on the need of including further information on the data gaps and homogeneities found in the different variables. We
have included a new table in the manuscript indicating these issues (please see below).

Section 2.3 L160-171 I do like the simple yet illustrate way to determine the effects of single variables on the detected trend. By design this is done as a local sensitivity analysis where one variable is changed holding the others fixed. However, it is not a global sensitivity analysis and co-variation of the forcing variables is neglected. Especially for the meteorological variables used here, I suspect that the variables and eventually their trends in time do co-vary - e.g. temperature and relative humidity. Did you consider such effects and are they important to understand the long-term variability?

We agree that co-variation of the forcing variables is not considered here because co-variation between meteorological variables in the Canary Islands is really low. This is illustrated in a set of box-plots below in which the correlation between the seasonal and annual series of the meteorological variables in the eight meteorological stations is shown. With the exception of the high positive correlations found between maximum and minimum air temperatures, the correlation among the other variables is really low and mostly non-significant. Only in winter and spring there are dominant significant correlations between the sunshine duration and the maximum air temperature. Given the strong independence in the variability of the different climate variables the co-variation study suggested by Reviewer#1 would not provide any new result in comparison with that applied here. In any case, we much appreciate this raised comment.

L195 . . . the aerodynamic component (Eq. 3).

Replaced.

Discussion: L303- 305 differences in ET0 trends across sites . . . "must be considered either due to random effects and uncertainty at various levels or due to microgeographic effects . . . " I think the differences of the trends and the different strength of the aerodynamic / radiative components at the sites deserve more attention in the discussion. The results are presented already in a detailed manner and these aspects should be discussed. That means is there a link of the different strength of certain

forcing variables and the magnitude of ET0 trend at a given site.

We have included some discussion about this issue in the revised manuscript:

"There is not a general pattern that may connect the observed trends in a certain forcing variable with the observed trend of ETo in each of the eight analysed stations although those that showed a higher increase in ETo (i.e., Lanzarote, Los Rodeos and Gran Canaria) displayed a higher increase in the aerodynamic component; a process which is in agreement with the significant reductions observed in relative humidity."

L514: in preparation

Replaced in the revised manuscript.

L195 . . . the aerodynamic component (Eq. 3).

Replaced.

Discussion: L303- 305 differences in ET0 trends across sites . . . "must be considered either due to random effects and uncertainty at various levels or due to microgeographic effects . . . " I think the differences of the trends and the different strength of the aerodynamic / radiative components at the sites deserve more attention in the discussion. The results are presented already in a detailed manner and these aspects should be discussed. That means is there a link of the different strength of certain forcing variables and the magnitude of ET0 trend at a given site.

We have included some discussion about this issue in the revised manuscript:

"There is not a general pattern that may connect the observed trends in a certain forcing variable with the observed trend of ETo in each of the eight analysed stations although those that showed a higher increase in ETo (i.e., Lanzarote, Los Rodeos and Gran Canaria) displayed a higher increase in the aerodynamic component; a process which is in agreement with the significant reductions observed in relative humidity."

L514: in preparation

Replaced in the revised manuscript.

Figure 1: Please add LAT - LON coordinates as a grid and a scale for the distance

Figure 1 has been replaced following the Reviewer's suggestion.

Figure 4 and 5: The grey lines are not very informative. Please adapt the figures, using different colors or line types for the sites. It might be also useful to demean the time series for the display. In the annual panel the bold line is missing.

We have removed grey lines and only included the two meteorological stations with longest records (Izaña in green, and Santa Cruz de Tenerife in brown)

Fig. 5 the labels are too small to be readable

Labels have been replaced to be readable.

Finally, we wish to thank the Reviewer#1 for reviewing our paper and for your useful comments/suggestions.

———————————————

| Code | Longitude | Latitude | Name | relocation | Relative humidity | | Sunshine duration | | Wind speed | | maximum temperature | | minimum temperature | |
|---|---|---|---|---|---|---|---|---|---|---|---|---|---|---|
| | | | | | data gaps | Inhom. | data gaps | Inhom. | data gaps | Inhom. | data gaps | Inhom. | data gaps | Inhom. |
| C0290 | -13.60 | 28.95 | Lanzarote/Airport | 1972 | 2.20% | 1998 | 0.78% | 1978-2002 | 0.47% | 1971 | 1.23% | 2004 | 1.23% | 1988 |
| C139E | -17.75 | 28.61 | La Palma/Airport | 1970 | 0.94% | | 2.51% | | 0.47% | 1976 | 0.37% | | 0.37% | 1997 |
| C249I | -13.85 | 28.43 | Fuerteventura/Airport | 1969 | 0.15% | 2000 | 1.25% | 1995-2005 | 0.15% | | 0.23% | 1983 | 0.23% | 1977 |
| C430E | -16.48 | 28.30 | Izaña | -- | 1.72% | 1999 | 7.40% | 2005 | 6.91% | | 5.20% | 1985 | 5.20% | |
| C447A | -16.31 | 28.46 | Los Rodeos | -- | 0.31% | | 1.10% | 1966 | 0.15% | 1970 | 0.30% | 2005 | 0.30% | 2005 |
| C449C | -16.25 | 28.45 | Santa Cruz de Tenerife | -- | 0% | | 0.94% | | 0% | 1987 | 0% | | 0% | 1994 |
| C649I | -15.38 | 27.91 | Gran canaria/Airport | -- | 0.15% | 1981-1994 | 2.67% | 1978 | 0.31% | 1972 | 0.20% | 1984 | 0.20% | 1994 |
| C659P | -15.41 | 28.15 | San Cristóbal | 1994 | 11% | | 1.88% | 1980 | 10.50% | 1994 | 5.30% | 1966 | 5.30% | |

**Fig. 1.**

[Figure]

**Fig. 2.**

---

## Author Comment (AC2) · 25 May 2016

Interactive comment on "Recent changes and drivers of the atmospheric evaporative demand in the Canary Islands" by S. M. Vicente-Serrano et al. Anonymous Referee #2

The manuscript deals with an analysis of the atmospheric evaporative demand (AED) over the Canarian Island for the period 1961-2013. Basis are meteorological data (monthly, p4l96) from 8 stations which are used as inputs for the FAO-56 Penman-Monteith equation to derive monthly AED. While the paper is generally well written, I feel there are a number of conceptuals issues that need to be resolved and addressed before a possible acceptance.

Thanks for your positive assessment of our manuscript and strong effort providing constructive comments. Addressing your comments has helped improved our revisions.

Please find below our answers to each comment and if you have any residual concerns please feel free to raise those points.

- As the FAO-56 is a non-linear equation that has been developed for daily inputs, how do authors justify the application of monthly average input values?

The FAO-56 equation can be obtained from daily and monthly records, as Allen et al., 1998 stated: "the FAO Penman-Monteith equation requires air temperature, humidity, radiation and wind speed data for daily, weekly, ten-day or monthly calculations". Several previous studies focused on drought using ETo at larger spatial scales have also computed the Penman Monteith ETo using monthly values for some variables. Using monthly averages instead of higher temporal resolution of data (e.g., daily) has not a relevant influence on the ETo estimations. An example is showed below for the ETo calculations in two stations of the Canary Islands (Los Rodeos and Izaña) for the 1978-2010 period based on the raw climatic data. The figure shows the relationship between the monthly ETo sum from daily measurements and the calculations from the average of monthly climate variables, which justifies the equality of applying both procedures. This is clearly observed for the ETo monthly values (including seasonality) but also considering monthly standardized anomalies in which seasonality is removed. In addition, we would like to remark that obtaining high quality and homogeneous time series of the necessary variables for calculating ETo on a daily basis is highly problematic since there are not robust methodologies to homogenize climate variables at the daily scale, whereas testing and correcting homogeneity using monthly records is reliable. Given high agreement of both daily and monthly ETo estimations but stronger robustness and homogeneity of monthly series, it seems recommendable to use monthly climate records in climate change studies.

- As some of the input variables (Rn) have to be estimated from other parameters, some of the discussion about these relationships (p.14) need to be provided earlier in the text.

The estimation of the Rn is the exception in relation to the other variables needed to apply the FAO-56 Penman-Monteith equation. Rn is indirectly estimated by means of sunshine duration records. We agree with the Reviewer#2 and the implications of this method are discussed in depth in the discussion section. Furthermore, in the revised manuscript we have moved some of this discussion to the section of methods.

- While in general there are many graphical illustration for plenty of aspects, I actually miss graphs with the temporal dynamics and developments of input variables into the FAO-56 equation. Where can I see the trend for Rn, T, wind speed, rH? This would be important as they are the controlling variable in the equation.

This is included in Table 7 (Table 8 in the revised manuscript). This table shows the magnitude of change for air temperature, relative humidity, sunshine duration and wind speed in $°C$, %, hours and m s-1 decade-1, respectively, over the 1961-2013 period. This is analysed for the different available meteorological stations but also for the regional series. In addition, Table 8 also includes the statistical significance of the observed changes at the confidence 95% confidence level.

- Why are authors relating calculated ET0 with variables that have been used to calculate ET0 before (or used to derive inputs from where ET0 is calculated) - see for example Fig. 6. Why don't authors simply calculate the sensitivities (partial derivatives) of FAO-56 with respect to the driving variables. I simply did that and only from using a temperature increase of 0.6 âŮęC (keeping specific water content constant) and some realistic Rn, T , ra, rs – values (I used the original PM formula) I could derive the changes in ET0 stated by the authors. I feel a sensitivity study in this way including trend analysis of the inputs would be more compact and informative for the readers.

In the manuscript we already combined these two suggested approaches. On the one hand, we determined the relationship between ETo calculations and the interannual variability of the different meteorological variables; on the other hand, we also followed the approach using the PM equation, including trend analysis. This was explained in

the methods section:

"...we applied the PM equation while holding one variable as stationary (using the average from 1961 to 2013) each time. This approach provided five simulated series of ETo, one per input variable, which could be compared to the ETo series computed with all the data to determine the isolated influence of the five variables. Significant differences between each pair of ETo series (i.e., the original one and the alternative one in which one variable was kept constant) were assessed by comparing the slopes of the linear models, with time as the independent variable. A statistical test for the equality of regression coefficients was used following Paternoster et al. (1998). The significance of the difference was assessed at a 95% confidence level (p<0.05)."

Figures 7 and 8 show the results of this analysis.

- Authors state they applied the Mann-Kendall – did they check and correct for autocorrelation?

We considered the autocorrelation in the trend analysis applied to the series of (i) ETo, (ii) aerodynamic and radiative components of the ETo, and (iii) the series of the different climate variables (i.e., air temperature, relative humidity, wind speed and sunshine duration). This was applied using the FUME R package, which performs the modified Mann-Kendall trend test, returning the corrected p-values after accounting for temporal pseudorreplication (Hamed and Rao, 1998; Ye and Wang, 2004). This has been detailed in the revised manuscript.

Hamed, K.H. and A.R. Rao, (1998). A modified Mann Kendall trend test for autocorrelated data. Journal of Hydrology 204, 182-196. Yue, S. and C. Wang (2004). The Mann-Kendall Test Modified by Effective Sample Size to Detect Trend in Serially Correlated Hydrological Series. Water Resources Management 18, 201-218.

Overall, I feel there is still a large potential to improve the overall structure/concept of the manuscript as outlined above. As a result I suggest major revisions to the

manuscript before publication.

Finally, we thank Reviewer#2 for the revision task and the useful comments raised for impro ving the results presented in this manuscript. Hopefully we have answered to all these major and minor concerns satisfyingly; otherwise we are available for further clarifications.

[Figure]

**Observed ETo**

Los Rodeos

$R^2 = 0.99$

Monthly ETo

Monthly sum of daily ETo

Izaña

$R^2 = 0.99$

Monthly ETo

Monthly sum of daily ETo

**St. Anomalies**

Los Rodeos

$R^2 = 0.99$

Monthly ETo

Monthly sum of daily ETo

Izaña

$R^2 = 0.96$

Monthly ETo

Monthly sum of daily ETo

**Fig. 1.**

---

## Author Response (AR2)

Dear Dr. Pfister:

Please find attached a revision of the manuscript entitled "Recent changes and drivers of the atmospheric evaporative demand in the Canary Islands" to be considered for publication in Hydrology and Earth System Sciences. In the revised manuscript, we have addressed all comments and suggestions provided by the reviewer. You will also find enclosed a letter that includes a detailed response to his/her comments.

We look forward to hearing from you at your earliest convenience, and if you have any questions please feel free to contact me.

Sincerely,

Sergio M. Vicente-Serrano and co-authors.

**Following specific comments made by one referee, there now remain only some minor edits that need to be made to your manuscript. Before final publication, you would have to focus on a few specific points:**

**- develop in the manuscript on the non-linearity of the PM equation**

We have included the following new information in the revised manuscript:

*"The FAO-56 PM is an equation initially designed for crop monitoring and irrigation operation at daily and sub-daily scales. This equation involves non-linear relationships among the variables used for calculation and averaging these variables for long-term intervals could affect the reliability of the ETo estimations. Nevertheless, Allen et al. (1998) indicated that the FAO-56 PM equation can be used for daily, weekly, ten-day or monthly calculations, and several previous studies have computed the Penman Monteith ETo using monthly values for some variables (e.g., Sheffield et al., 2012; Dai, 2013). We have found that using monthly averages instead of daily records for the different variables has not a relevant influence on the ETo estimations in the Canary Islands. Figure 2 shows an example using two of the available stations (Los Rodeos and Izaña) for the 1978-2010 period. The relationship between the monthly sum of the daily ETo calculations and the ETo calculation from the monthly averages, justifies the equality of applying both procedures. This is observed for the ETo monthly values (including seasonality) but also considering monthly standardized anomalies in which seasonality is removed. Moreover, there are other technical reasons that recommend the use of monthly instead daily records to calcule ETo since testing and correcting the temporal homogeneity of the necessary variables on a daily basis is highly problematic, whereas testing and correcting homogeneity using monthly records is reliable (e.g. Venema et al., 2012)."*

[Figure]

Figure 2. Comparison between the average monthly ETo obtained from daily meteorological records and the ETo directly calculated from monthly meteorological variables. Two meteorological stations in the Canary Islands are used for the period 1978-2010 (Los Rodeos and Izaña). The figure shows the relationship between monthly ETo series but also between the series of standardized anomalies in which seasonally is removed.

**- improve the figure captions for figs. 7 and 8**

We have changed the figure captions for figs. 7 and 8 (Figures 8 and 9 in the revised version):

Figure 8: Relationship between the observed change in ETo (mm. year-1) in each meteorological station and the change in simulated ETo considering each one of the meteorological variables used to calculate ETo as constant for the period 1961-2013. Black dots indicate significant differences in the trends.

Figure 9: Seasonal and annual evolution of the observed regional ETo compared to the evolution of simulated ETo considering each one of the meteorological variables used to calculate ETo as constant for the period 1961-2013.

---

## Author Response (AR3)

Dear Dr. Pfister:

Please find attached a revision of the manuscript entitled "Recent changes and drivers of the atmospheric evaporative demand in the Canary Islands" to be considered for publication in Hydrology and Earth System Sciences. In the revised manuscript, we have corrected the small typo you indicated.

We look forward to hearing from you at your earliest convenience, and if you have any questions please feel free to contact me.

Sincerely,

Sergio M. Vicente-Serrano and co-authors.